# Effects of Metal Waste Strips on Strength Characteristics of Road Base Material

**Osama Ragab Ibrahim** *, **Mazoun Al Sinani**, **Israa Al Sinani**, **Bayan Al Shibli**, **Amjaad Al Badi** and **Salma Al Maghawry**

Faculty of Engineering, Sohar University, Sohar P.O. Box 311, Oman; 180156@students.su.edu.om (M.A.S.); 180170@students.su.edu.om (I.A.S.); 180175@students.su.edu.om (B.A.S.); 180306@students.su.edu.om (A.A.B.); 181291@students.su.edu.om (S.A.M.)
* Correspondence: oibrahim@su.edu.om

**Abstract:** Metal manufacturing produces various types of byproducts and metal waste that have been growing exponentially. The increasing amounts of metal waste are usually disposed of in landfills, which causes soil and water pollution and increases the amount of gas emissions. On the other hand, in the field of pavement construction, the demand for natural materials is increasing rather rapidly. Many studies have suggested utilizing aggregate-like waste material in pavement construction, but there is little to no literature documenting the use of metal strips of different types. The aim of this study is to investigate utilizing the metal waste produced by the Intag Sohar company in layers of flexible pavement. Selected types of metal waste were utilized in the construction of the material used for base and subbase road layers. Three main parameters were studied: the number of layers, the distance between strips, and the direction of the strips. The effect of the metal waste on the pavement material is evaluated using the standard California Bearing Ratio test (CBR), which is the most important indication of the strength of the pavement material. Results show that the highest-quality mix consisted of four layers of metal waste with 1 cm spacing in one direction with CBR% values equal to 118.807%.

**Keywords:** waste; sustainable construction materials; subbase course; base course; California bearing ratio

## 1. Introduction

Since ancient times, humans have depended on movement and transportation to be able to meet their needs, so highway construction is important to facilitate human transportation [1]. Moreover, the development and progress of any country can be seen through the way it designs highways [2]. Currently, due to the rather rapid increase in population growth, rapid economic development, and urbanization, there has been an increase in traffic load and traffic volume. Generally, the increase in traffic load has led to lower pavement performance [3]. Therefore, it can be observed that developed countries pay a great deal of attention to highway construction [4]. Hence, soil stabilizing has been introduced in various civil engineering projects. Soil stabilizing is the process of enhancing the engineering properties of soil to make it more stable and durable. The application of soil stabilizing in highways and pavements assists road engineers in implementing materials of certain high mechanical properties to meet the needs of highways pavements.

Typically, the design of flexible pavement consists of three main layers beginning with the surface course. The surface course is the top layer and serves as a moderately impermeable course that is robust, smooth, and abrasion resistant [5]. It must withstand the imposed wheel loads and safely transmit them to the layer below, since it is in close contact with the tires of the vehicle [6,7]. The base course, which is built below the surface course, equally disperses the loads passed via the surface course onto the layers below [6]. Next is the subbase course, which is immediately below the base course and helps the courses

above it to distribute the loads more effectively [8]. It may be made up of soil that has been stabilized or soil aggregate mixtures that let free water drain off the pavement [9]. The lowest layer of the flexible pavement is the subgrade layer. Which serves as the highway's foundation [10]. If any layer is not of sufficient strength, then stabilization and enhancement are required [11].

Industrial development leads to a direct increase in the industrial waste-generating rate. It disrupts the ecological cycle and contaminates the nearby bodies of water and soil. The use of waste from various materials, such as iron, aluminum, and others to strengthen the road has several benefits from an economic viewpoint [12,13]. Utilization of various types of waste is an efficient way to address recycling issues while simultaneously improving the performance of the pavement system [14]. When the layers of the road are strengthened, they last for a longer period than usual without the need for continuous maintenance, meaning that the durability of the pavement significantly increases [15,16]. Therefore, officials will not need to pay other sums for road maintenance since the waste material strengthens the pavement against deformation and increases the lifespan of the pavement [17]. Moreover, it reduces the cost of materials required for the pavement, reduces the demand on landfill areas, and decreases the transportation fees of these waste materials. Waste constitutes an increasing threat to the environment, which leads to basic environmental problems, most notably pollution of all kinds [18,19]. Therefore, it is assumed that these wastes will be disposed of in the most environmentally conscious way, or that they will be exploited effectively and beneficially [20,21].

Recently, there have been several research studies promoting the use of various waste materials in pavement construction. Kabeta studied the effectiveness of adding plastic strips to weak clay soil. The samples were assessed using the unconfined compression test (UCS), the California bearing ratio (CBR), and the compaction test. Results showed that the strength of the clay improved significantly after incorporating plastic strips [22]. Kassa et al. also utilized plastic strips as stabilizing material for expansive soil with ratios of 0.5%, 1% and 2%. Similar results were found in that there was an improvement in terms of the swelling and cracking of the soil [23]. Moreover, similar findings were obtained by Saravanan et al. when using waste synthetic strips with ratios of 0.5%, 1%, and 1.5%. Their findings demonstrated that the expanding behavior of clayey soil was reduced [24]. Overall, outcomes encourage the use of plastic products as stabilizers for soils [24].

The use of the Electric Arc Furnace (EAF) Slag was also investigated by studying its volume expansion. Sukmak et al. found that soaking EAF in 2 wt.% acetic acid provides the minimum volume expansion and that it is encouraged for use of EAF in road applications [25]. Kontoni et al. studied the employment of gene expression programming (GEP) in generating models that predict the behavior of sustainable construction material. Such use of machine learning techniques leads to innovative solutions in sustainable construction applications [26]. Reviews have also been conducted on the use of mining waste, including waste rock and mine tailings, in civil applications as a way of reducing the effect of mining waste on the environment and preserving natural resources [27].

In this study, the main aim is to introduce soil stabilizing via the sustainability of the metal waste obtained from the Intag Sohar Company, which specializes in shaping and carving metal molds at Sohar University, for pavement applications. Three types of metal strips (carbon steel, engineering teel, and stainless steel) were used at the base of the roads to reinforce and increase the strength of the road layers, thus the negative impact of the metal waste on the environment will be limited [28,29]. The optimum mixture of the natural material and the waste metal strips was obtained by studying three main parameters: the number of layers, the distance between strips, and the direction of the strips. The effectiveness of each mixture was examined through experimental examination, mainly through the use of the California bearing ratio (CBR) test [30].

This study is unique in that it is one of the few studies conducted regarding the use of waste metal strips in the Middle East. The utilization of waste metal strips will lead to a significant decrease in the demand for landfill areas and prevent said areas' negative

impacts on the environment. This is especially significant for groundwater, which serves as the main water resource in Oman. Moreover, this study will help preserve the natural resources that are currently facing depletion, since they are being used for all construction applications. Overall, this study will increase the lifetime, durability, and serviceability of the roads in an economically and environmentally friendly way.

## 2. Materials and Methods

The material that was used in this research was collected from Intag Sohar company. The Intag Sohar project is considered one of the most important industrial projects specializing in the formation and carving of metal molds of all kinds. The company specializes in advanced manufacturing of metals, which results in the processes of sculpting and forming thin metal waste that comes in different shapes and sizes. There are short strips, long strips, and slices, and each of them differs according to the material of the formed or carved mold, as the mold itself differs in its components and chemicals. Mineral waste strips were taken from Intag Sohar and used as road layers.

The chemical components of the three waste metals are shown in Table 1. The type used for this study is carbon steel grade 1026 (C-1026) the mechanical properties of which are shown in Table 2, engineering steel (EN-08), the mechanical properties of which are shown in Table 3, and stainless steel (S316). Waste metal strips are shown in Figure 1, while the road base material used is shown in Figure 2. This type of waste material has excellent corrosion resistance, the ability to maintain a clean surface, higher creep, higher stress-to-rupture, high durability, increased formability, and tensile strength at elevated temperatures.

**Table 1.** Chemical Components of C-1026, EN-08, and S316.

| Component | Weight% | | |
| :---: | :---: | :---: | :---: |
| | C-1026 | EN-08 | S316 |
| C | Max 0.28 | 0.36–0.44 | Max 0.08 |
| Fe | 98.73–99.18 | - | - |
| Mn | 0.60–0.90 | 0.60–1.0 | - |
| P | Max 0.040 | Max 0.050 | 0.045 |
| S | Max 0.050 | Max 0.050 | - |
| Si | - | 0.10–0.40 | 0.75 |
| Mg | - | - | Max 2 |
| Ni | - | - | 10–14 |
| N | - | - | 0.10 |
| Ag | - | - | 0.030 |

**Table 2.** Mechanical Properties of C-1026.

| Mechanical Properties | Metric | Comments |
| :---: | :---: | :---: |
| Hardness, Brinell | 143 | - |
| Hardness, Knoop | 163 | Converted from Brinell hardness. |
| Hardness, Rockwell b | 78 | Converted from Brinell hardness. |
| Hardness, Vickers | 149 | Converted from Brinell hardness. |
| Tensile Strength, Ultimate | 490 Mpa | - |
| Tensile Strength, yield | 415 Mpa | - |
| Elongation At Break | 15% | In 50 Mm |
| Reduction Of Area | 40% | - |
| Modulus Of Elasticity | 205 GPa | Typical For Steel |
| Bulk Modulus | 160 Mpa | Typical For Steel |
| Poisson's Ratio | 0.29 | Typical For Steel |

**Table 3.** Mechanical Properties of EN-08.

| Mechanical Properties | Metric | Comments |
|---|---|---|
| Max stress | 700–850 Mpa | - |
| yield Stress | Min 465 Mpa | Up to 19 mm LRS |
| 0.2% Roof Stress | 450 Mpa | Up to 19 mm LRS |
| Elongation | 16% Min | 12% if cold drawn |
| Impact KCV | 28 Joules Min | Up to 19 mm LRS |
| Hardness, Brinell | 201–255 | - |

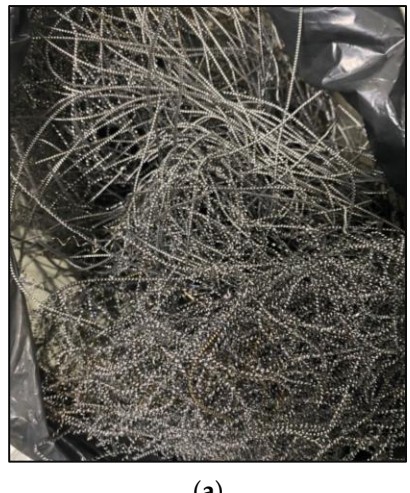
(**a**)

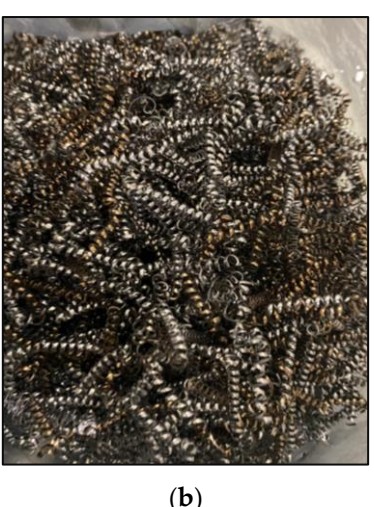
(**b**)

**Figure 1.** Waste metal strips used in this study were obtained from Intag Sohar company. (**a**) Stainless steel (S316); (**b**) A mix of Carbon steel (C-1026) and Engineering steel (EN-08).

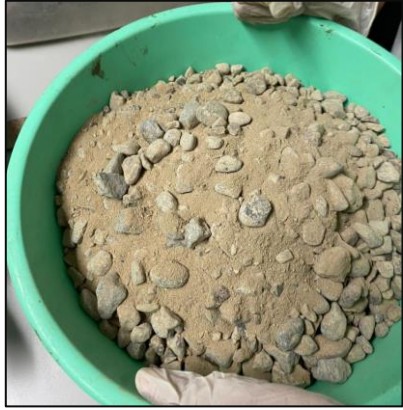

**Figure 2.** The road base material used in this study.

Table 1 shows that stainless steel (S316) contains more elements in its chemical composition than the other two types of steel, being composed of varying percentages of silicon, magnesium, nickel, silver, and nitrogen. The mechanical properties of the metal strips listed in Table 3 show that engineering steel (EN-08) has a maximum stress of 700–850 Mpa while Carbon steel (C-1026) has a maximum stress of 490 Mpa, which indicates that EN-08 has more resistance than C-1026. Moreover, the Brinell Hardness values of EN-08 and C-1026 are 201–255 and 143, respectively, which also indicates that EN-08 has more hardness than C-1026.

First, a soil sample was taken from an area in Sohar in order to study the mechanical properties of the natural soil and assess its ability to bear high loads using the CBR test and use the values for the natural soil as a reference point for the strength after adding

the waste materials from Intag Sohar company. Two types of steel strips were studied on natural soils, and for each type, the study was conducted based on three parameters. The first parameter is the distance between the strips, where three distances were taken: 0.5 cm, 1 cm, and 1.5 cm. The second parameter is the number of layers that can be placed between the soil to increase its bearing capacity, and these tests were conducted on five layers. The last parameter is the direction of the strips and the effects of placing them in one direction versus two directions. Figure 3 shows the flowchart of the methodology of the study.

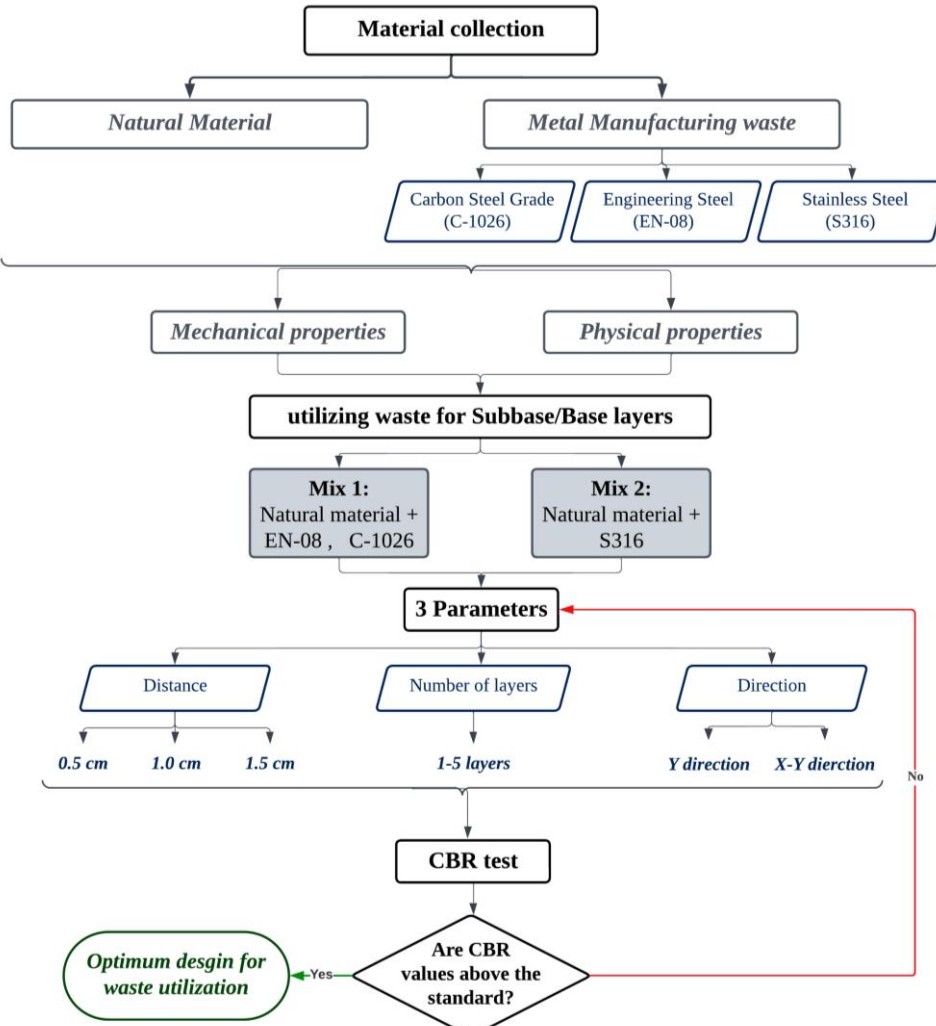

**Figure 3.** Flowchart of the methodology of the study.

## 3. Results and Discussion

### 3.1. Natural Material

The mechanical properties of different soil samples collected from Sohar were analyzed and determined using various tests, including the grain sieve analysis test, the liquid limit (LL), the plastic limit (PL), and the plastic index (PI) according to the Atterberg limit test. Subsequently, the most suitable sample that best conforms to the standards that apply to the base layer of the road is selected. The results of the particle size distribution test of the natural material are shown in Figure 4, the laboratory results of the proctor test are shown in Figure 5, and the results of the mechanical and physical properties of the material are shown in Table 4. The gradation of soil is found to be within the gradation limits of the base material according to American Association of State Highway and Transportation Officials (AASHTO) standards. Using the particle size distribution, the uniformity coefficient (Cu) and the coefficient of gradation (Cc) are calculated. The uniformity coefficient is an

expression of the variation of sizes of the particles in the natural soil, while the coefficient of gradation expresses the degree of soil gradation. The values of Cu and Cc were found to be equal to 65 and 1.3, respectively. Such values indicate that the natural soil is well-graded.

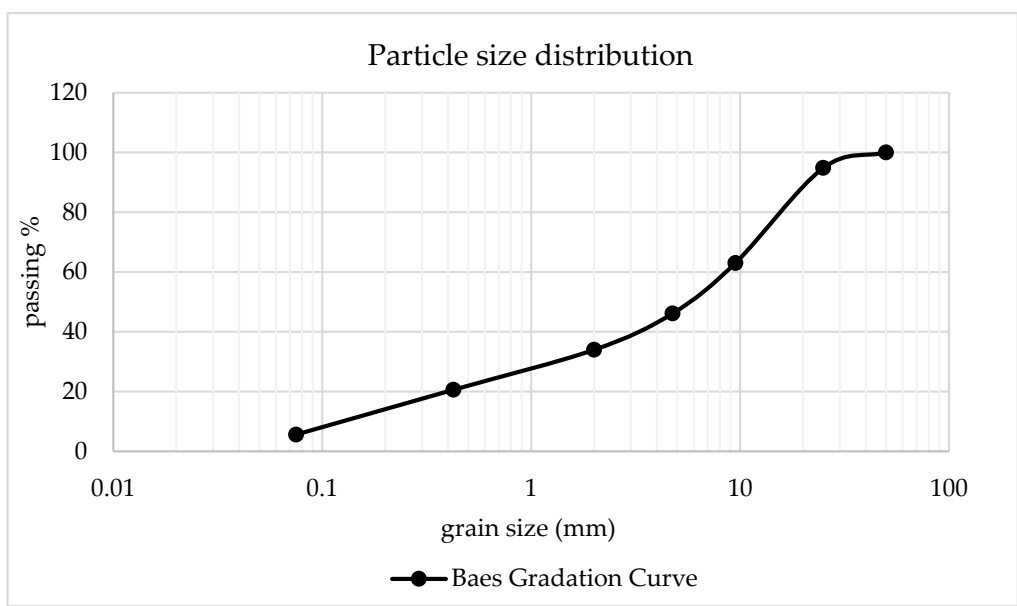

**Figure 4.** Particle size distribution of the natural Base material.

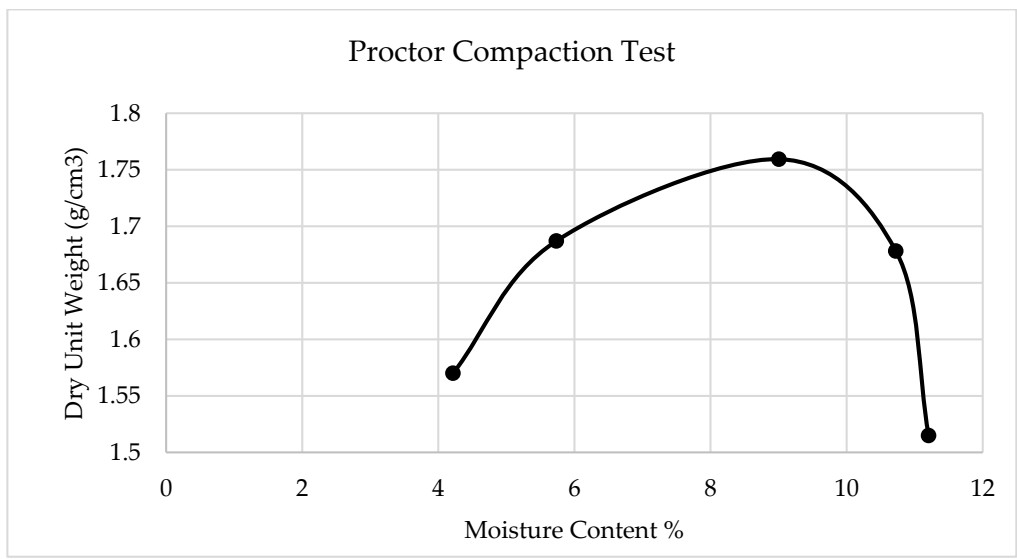

**Figure 5.** Proctor Compaction Test for the natural Base material.

**Table 4.** Mechanical properties of the natural base material.

| Test | Standard Values | Result |
|---|---|---|
| Liquid limit | 2% Max | 23.82% |
| Plastic limit | - | 18.11% |
| Plastic index | 6% Max | 5.82% |
| Optimum moisture content (OMC) | - | 9% |
| Maximum dry density | - | 1.76 g/m$^3$ |
| Los Angles (abrasion value) | 43% Max | 12.87% |
| Impact value | 10–20% | 13.13% |

In terms of the plastic and liquid behavior of the material, the water content at which the behavior of clayey soil changes from plastic to liquid was determined. The liquid limit (% LL = 23.82) is less than the maximum value of the base layer standard (25%), therefore it is acceptable. Moreover, the average water content of plastic limit was found to be equal to 18.12%. Subsequently, the Plasticity Index (PI) was found to be equal to 5.82%. Since 5.82% < 6%, which is the maximum value of the base layer, the soil is considered acceptable for use as a base layer. As for the compaction test of the material, it was found that the dry density increases the moisture content until it reaches optimum moisture content. Subsequently, the dry density decreases as moisture content increases. After analyzing the results, the maximum dry density was found to be equal to 1.76 g/m$^3$, and the optimum moisture content was found to be equal to 9%, which will be used for the preparation of the sample for the CBR test.

The abrasion value of the aggregate represents the percentage of wear due to relative rubbing action between the aggregate and steel balls. It was found to be equal to 12.8686%. Since the max permissible abrasion value ranges from 30–60%, the tested aggregate is acceptable and can be used in any type of pavement construction. Furthermore, the relative resistance to the impact of coarse aggregates was determined to be between 10–20%. Therefore, the tested aggregate sample is strong enough for pavement construction and it can carry the load without causing the pavement structure to fail. A CBR Test is used to evaluate the sample strength. The test setup of the CBR machine is shown in Figure 6. Where 100% represents the standard material (crushed limestone), the natural material CBR values were 78.637% and 80.177%, respectively. The CBR values of the natural material proves that this sample can be used as a base course layer.

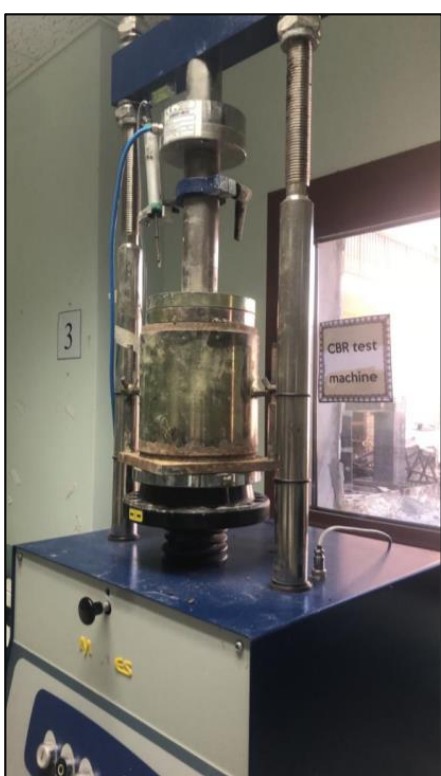

**Figure 6.** The CBR machine.

### 3.2. Waste Material Mixtures

3.2.1. Mixture 1: (Natural Material + EN-08, C-1026)

The results of the CBR test for each parameter of Mixture 1 are shown in Table 5.

**Table 5.** CBR results from Mixture 1 of waste with natural material.

| Material Used | No. of Layers | Spacing (cm) | Direction | CBR% |
|---|---|---|---|---|
| Natural material | - | - | - | 78.637 |
| Natural material + EN-08, C-1026 | 1 | 1 | Y | 83.267 |
| Natural material + EN-08, C-1026 | 1 | 0.5 | Y | 78.910 |
| Natural material + EN-08, C-1026 | 1 | 1.5 | Y | 79.083 |
| Natural material + EN-08, C-1026 | 2 | 1 | Y | 93.109 |
| Natural material + EN-08, C-1026 | 3 | 1 | Y | 101.987 |
| Natural material + EN-08, C-1026 | 4 | 1 | Y | 118.807 |
| Natural material + EN-08, C-1026 | 5 | 1 | Y | 80.006 |
| Natural material + EN-08, C-1026 | 1 | 1 | X-Y | 81.632 |

To study the effectiveness of the waste from Intag Sohar, three basic parameters were investigated: spacing, the number of layers, and the direction of waste strips. These parameters are used to measure the effectiveness of this metal waste within the natural material and to measure how increasing or decreasing the amount of waste material affects the strength of the road. These factors will contribute significantly to the analysis and provide access to the optimal mixture using all effective parameters that give the sample greater strength to bear high wheel loads. First, the effect of spacing, which is shown in Figure 7, is compared to the CBR value of the natural material, which is 78.637%. The percentage increase for 0.5 cm spacing, 1 cm spacing, and 1.5 cm spacing was found to be 0.34%, 5.5%, and 5.3%, respectively. Therefore, the maximum increase was achieved when the steel waste was placed with 1 cm spacing, in which it was increased from 78.637% to 83.267%. Consequently, for Mixture 1 (Natural material +EN-08, C-1026) the optimum spacing is 1 cm for the highest CBR value of 83.267%, which will be used to determine the other parameters.

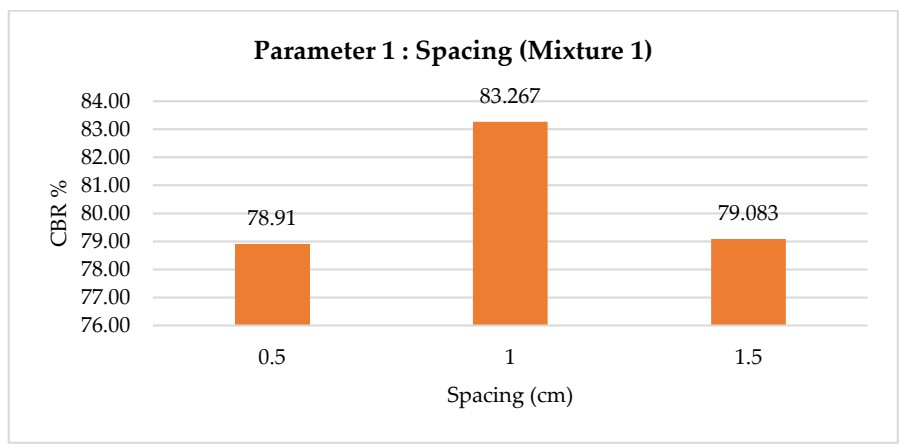

**Figure 7.** Comparing CBR results of Parameter 1: Spacing for Mixture 1: (Natural material + EN-08, C-1026).

The second parameter is the number of layers, as this material was gradually introduced into the base layer of the road (Figure 8). The results of the CBR test showed remarkable progress, and the strength of the road base increased significantly with the increase in the number of layers of waste material. A maximum strength increase of 16.5% was achieved at four layers. However, with the addition of a fifth layer of mineral waste to the mixture, a sudden drop of 32% occurred in the value of the CBR, bringing the total value of the CBR down to 80.006%. As a result, for Mixture 1 (Natural material + EN-08, C-1026) the optimum number of layers is four, with the highest CBR value of 118.807%. The CBR value of 118.807% shows that the sample has a significantly higher strength than the standard material (crushed limestone).

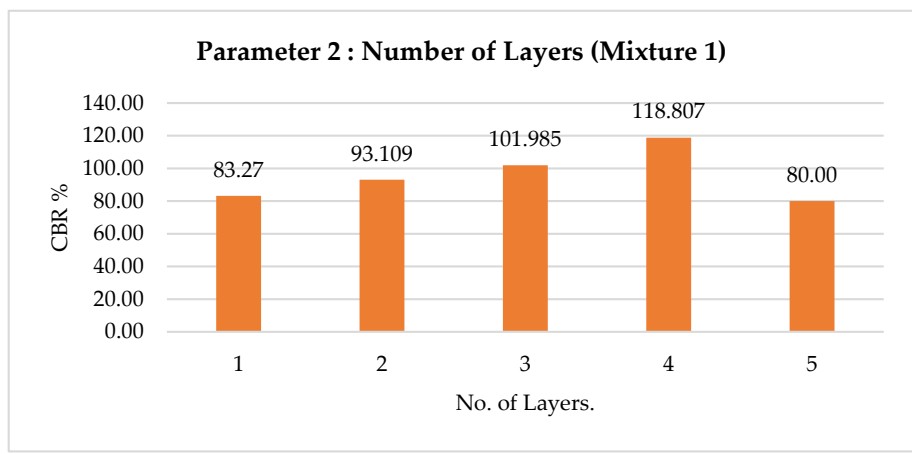

**Figure 8.** Comparing CBR results of Parameter 2: Number of Layers for Mixture 1: (Natural material + EN-08, C-1026).

The third parameter is the direction (Figure 9), as the strips were placed in one direction and then in two directions. Compared to the previous parameters the direction does not have such a significant effect. The lowest CBR value is equal to 81.632% for the mixture using one layer of the waste material with a spacing of 1 cm in two directions. Accordingly, for Mixture 1, placing the waste material in one direction results in a higher CBR value.

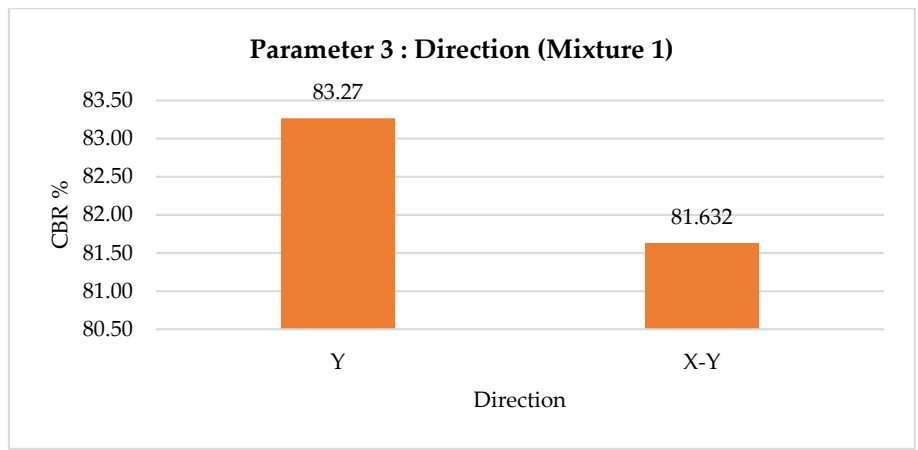

**Figure 9.** Comparing CBR results of Parameter 3: Direction for Mixture 1: (Natural material + EN-08, C-1026).

### 3.2.2. Mixture 2: (Natural Material + S316)

The results of the CBR test for each parameter of Mixture 2 are shown in Table 6.

**Table 6.** CBR results for mixture 2 of waste with natural material.

| Material Used | No. of Layers | Spacing (cm) | Direction | CBR% |
|---|---|---|---|---|
| Natural material | - | - | - | 80.177 |
| Natural material + S316 | 1 | 1 | Y | 81.730 |
| Natural material + S316 | 1 | 0.5 | Y | 80.954 |
| Natural material + S316 | 1 | 1.5 | Y | 80.399 |
| Natural material + S316 | 2 | 1 | Y | 83.812 |
| Natural material + S316 | 3 | 1 | Y | 84.267 |
| Natural material + S316 | 4 | 1 | Y | 84.967 |
| Natural material + S316 | 5 | 1 | Y | 85.350 |
| Natural material + S316 | 1 | 1 | X-Y | 82.175 |

For Mixture 2 (Natural material + S316), the spacing parameter (Figure 10) with spacing of 0.5 cm, the CBR value reached 80.9%, and then when the spacing distance was doubled to 1 cm, the CBR value reached 81.7%. In the third attempt, the spacing was increased to 1.5 cm, and the result was a decrease in the CBR value to 80.3%. Consequently, the optimal spacing, which can achieve the best result in increasing the strength of the base of the road, is between 0.8–1 cm, which results in a CBR value greater than 81.73%. This value will be used in studying the other parameters.

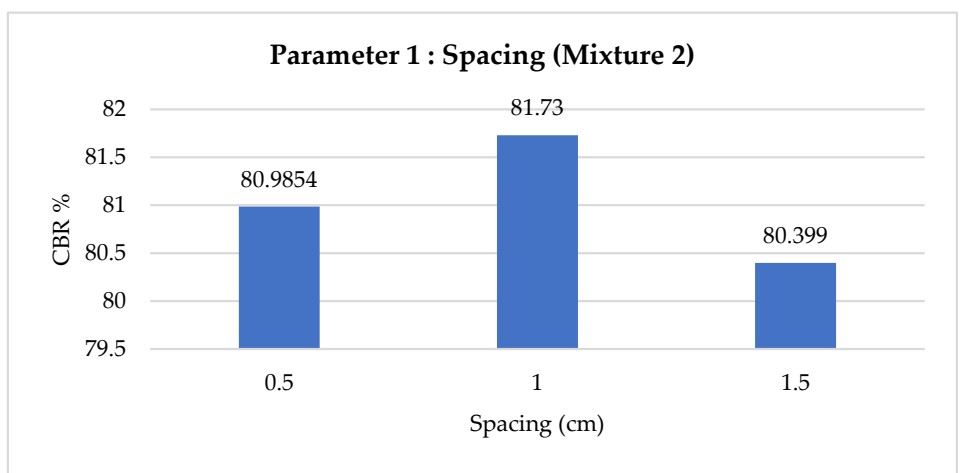

**Figure 10.** Comparing CBR results of Parameter 1: Spacing for Mixture 2: (Natural material + S316).

For the number of layers parameter, the results are shown in Figure 11, where the CBR value continually increased to reach its optimal state at the fifth layer, in which the CBR value was equal to 85.35. It is expected that the sample strength will continue to increase until a certain limit and then decrease, similar to what happened when testing Mixture 1. In the direction parameter (Figure 12), the results were higher when placing the waste in two directions as opposed to in one direction. Results show that the value of the CBR increased from 81.73% to 82.175%.

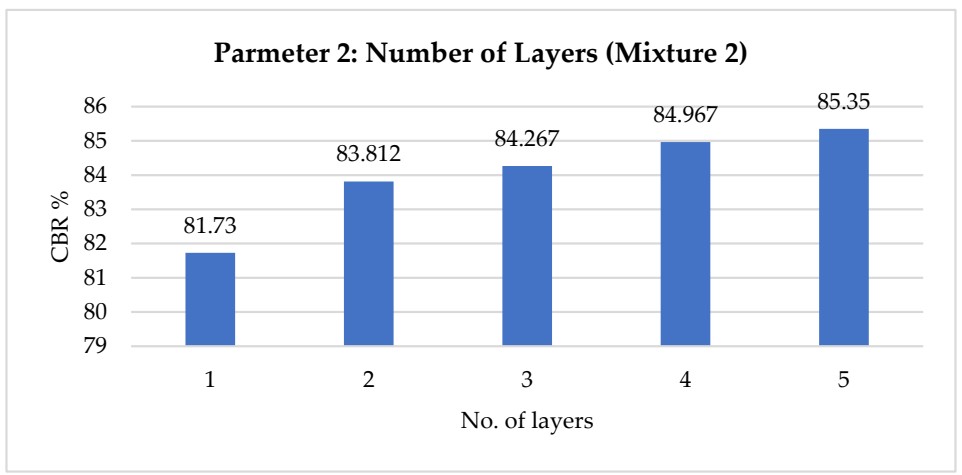

**Figure 11.** Comparing CBR results of Parameter 2: Number of Layers for Mixture 2: (Natural material + S316).

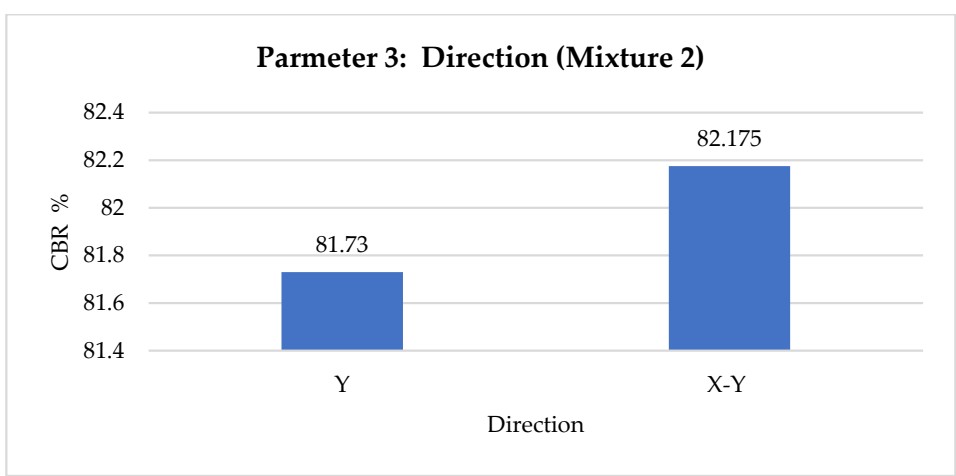

**Figure 12.** Comparing CBR results of Parameter 3: Direction for Mixture 2: (Natural material + S316).

*3.3. Comparing the Effects of the Two Mixtures*

It is known that each material contains specific properties that distinguish it from other materials. This is exactly what was found in this research, where each mixture of the waste material gave different results. A comparison between the performance of the two waste mixtures in terms of spacing, the number of layers, and the direction of placement is shown in Figures 13–15, respectively. Results show that the first waste material mixture was stronger than the second waste material mixture in terms of the spacing and layers parameters. It was found that for both mixtures, the spacing of 1 cm was the optimum spacing and showed the highest increase in the CBR value, However, Mixture 1 resulted in a CBR value of 83.26%, while Mixture 2 resulted in a CBR value of 81.73%, a difference of 1.85%. However, when metal strips are placed with a spacing of 0.5 cm or 1.5 cm, Mixture 2 performed better than Mixture 1, with an increase of 2.59% for 0.5 cm, which is the highest percentage for any spacing parameter, and 2.37% respectively.

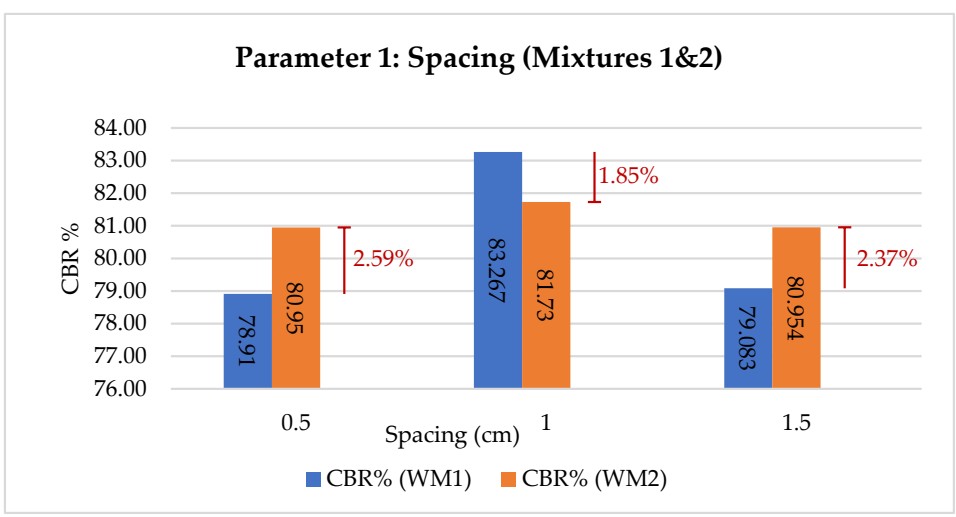

**Figure 13.** Comparing the two mixtures in the 1st parameter.

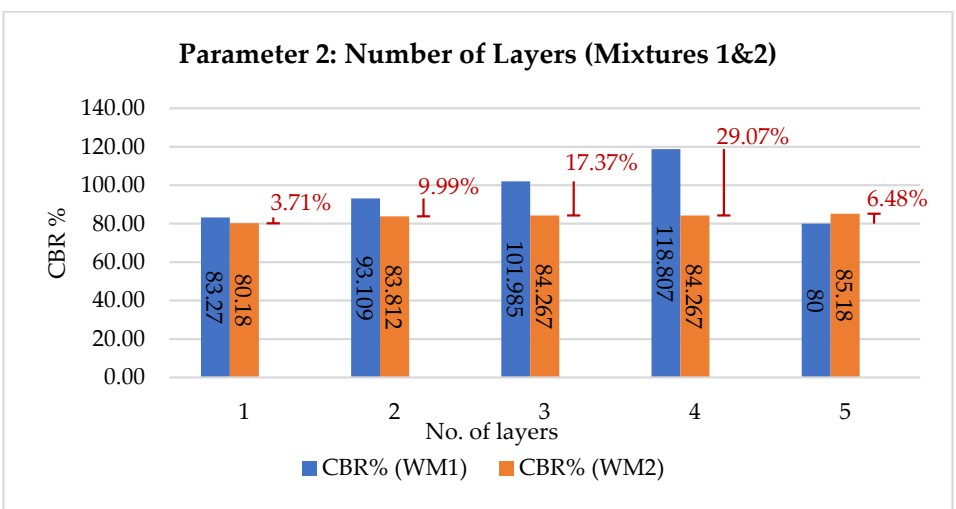

**Figure 14.** Comparing the two mixtures in the 2nd parameter.

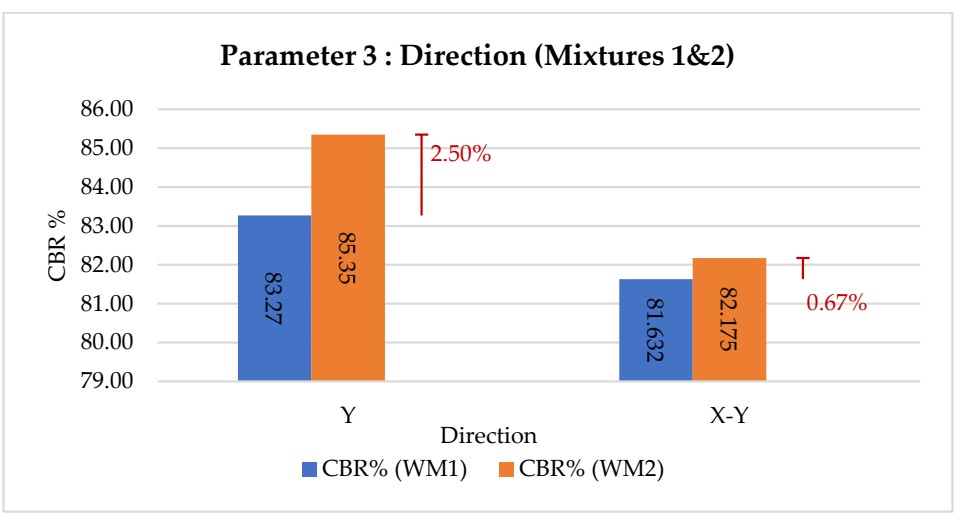

**Figure 15.** Comparing the two mixtures in the 3rd parameter.

In terms of the number of layers, Mixture 1 showed a significant increase when constructed with four layers, with a CBR value of 118.807%, which is 29.07% higher than Mixture 2. In contrast, the highest value for Mixture 2 was found at five layers with a value of only 85.18%. In nearly all cases, Mixture 1 resulted in CBR values higher than Mixture 2, with the exception of when five layers were used, in which Mixture 2 was 6.48% higher than Mixture 1. It is worth noting that the highest percentage difference between the two mixtures is recorded here, with the values of 17.37% for three layers and 29.07% for four layers., This shows that the number of layers parameter is the most significant difference between the two metal waste strips.

In terms of the third parameter, the direction of placement, there was a slight increase in Mixture 2. The placement of the material in one direction showed higher results, with a value of 85.35% compared to Mixture 1, which resulted in a value of 83.27%. Mixture 2 is 2.5% higher than Mixture 1, however, both mixtures performed better when only one direction was used for the placement of the metal strips. The percentage difference in the direction parameter is the lowest in comparison to the spacing and the number of layers parameters. As a result, the direction of the placement is the parameter that is the least significant between the two waste mixtures.

The strength of the base layer is one of the main input data for flexible pavement design. When the CBR value of the material is high, it indicates that the material has higher resistance to deformation, which results in a lower layer thickness in pavement construction. Consequently, the increase of the CBR of the material directly decreases the cost index of pavement construction. Moreover, the use of metal waste results in lower landfill area demand, and lower transportation costs of industrial waste. Overall, such results are crucial and beneficial to designing pavements that are economical, sustainable, and have sufficient strength and durability to withstand high wheel loads.

## 4. Conclusions

The current development of industrial activities has led to a significant increase in the generation rates of industrial waste. Metal manufacturing is one of the main industries that produce waste metal strips. Waste metal strips are disposed of in landfills and lead to contamination of the environment. This study was designed to investigate the potential of utilizing waste obtained from the process of manufacturing steel collected by Intag Sohar Company in road base material. Such an application will promote sustainable pavement, preserve natural resources, and increase the strength and durability of the pavement system.

First, the natural material from Sohar was ensured to be of sufficient strength to be used as a base layer according to the specifications of various tests. Three parameters were studied: the number of layers, the direction, and the spacing of the metal strips. The waste material was separated into two main mixtures, the first being Natural material + EN-08, C-1026 and the other being Natural material + S316. Each parameter was assessed using a CBR test. A CBR test is the main test indicating the strength of pavement material, where 100% represents the standard material (crushed limestone).

The obtained results indicate that the waste material increases the strength of the base layer when added in five layers, with 1 cm spacing, and placed in one direction. Overall, the first waste material is stronger than the second waste material based on their CBR values, in which the highest value of CBR obtained was 118.807%. It is recommended that further research be conducted to study the practical applications of the optimum mix that was obtained from this study. It is also recommended that future researchers use machine learning techniques to generate simulation models that predict the behavior of the pavement system using the obtained mixture. Statistical studies are also recommended using multiple trials of CBR testing, rather than only one trial for each parameter, as was completed in this study.

**Author Contributions:** Conceptualization, O.R.I.; methodology, M.A.S., I.A.S., B.A.S. and A.A.B.; investigation, O.R.I.; writing—original draft preparation, M.A.S., I.A.S., B.A.S. and A.A.B.; writing—review and editing, S.A.M.; supervision, O.R.I.; All authors have read and agreed to the published version of the manuscript.

**Funding:** This research received no external funding.

**Institutional Review Board Statement:** Not applicable.

**Informed Consent Statement:** Not applicable.

**Data Availability Statement:** No new data were created or analyzed in this study. Data sharing is not applicable to this article.

**Acknowledgments:** We acknowledge the contribution of the faculty of Engineering at Sohar University for providing the proper laboratory facilities to test the materials. Gratitude is also expressed towards the Department of Civil Engineering including professors and lab technicians for their assistance during the work of this research.

**Conflicts of Interest:** The authors declare no conflict of interest.

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
