# Peer review of "Effects of Metal Waste Strips on Strength Characteristics of Road Base Material"

_sustainability, doi:10.3390/su15129824_

Round 1

Reviewer 1 Report

Dear author(s).

Thank you for your efforts, the article is about the use of metal scraps as reinforcement material for the base layer. Although the paper is an applied and laboratory activity, it also has many problems, some of which are:

It is not necessary to mention the university name in the abstract. Be omitted.

- In the introduction, the necessity of conducting this research is not well defined, add explanations.

- In Figure 1, image B is not clear, change it.

- Tables 1, 3, and 5 can be combined and presented in one table.

-Why CBR is more than 100 for the base layer, usually the maximum of this variable is 80. Explain why this CBR value is.

 -In table 6, add the accepted standard values in a column.

-In figure 3, add the horizontal axis of the unit.

-In line 202 of page 7, the number of decimal digits and in other parts of the text and tables is sufficient, for example 13.13% is sufficient.

-In this article, three diagrams have been drawn for each of the materials and finally they were combined together, because the diagrams are simple, diagrams 4 to 9 should be removed (they are duplicates).

- This article has very little data. It is better that the respected authors provide more data from the laboratory.

 - The analysis and discussion about the reason for the changes in the factors examined in this study seem to be little and more explanation is needed.

thanks

The English text is fairly well written and seems acceptable.

Reviewer 2 Report

Is a good proposal to use of waste materials. I think is necessary point the number of samples for each test and present scatter bars in the graphics. 

Reviewer 3 Report

The following report is based on my review of the manuscript entitled “Experimental study on the effect of Metal waste strips on strength characteristic of road Base material”, with manuscript number “sustainability-2405415”. The manuscript fits within the scope of “sustainability” and is also interesting. However, the following shortcomings have been pointed out and need to be addressed properly for further improvement of the manuscript. They are:

1-     The title should be shortened. It is suggested to use: “Effect of metal waste strips on strength characteristic of road Base material”.

2-     Since all authors are from the same institution, the affiliation should be only one. It should not be 1-6. Kindly modify.

3-     The abstract needs to be modified. Rewrite the abstract completely as there are many mix-ups. It should be more specific and informative. It should contain Objectives, Methods/Analysis, Findings, and Novelty /Improvement Authors need to clearly state the motive for the research (i.e. problem statement). Try as much as possible to avoid examples in the abstract. “such as (Carbon Steel, Engineering Steel, and Stainless steel)”.

4-     There is a mixture of capital and small letters in the “title” and “keywords section”. Kindly make it uniform.

5-     So many paragraphs in the entire manuscript are either too lengthy or too short. Kindly consider at least 4 and at most 9 sentences in each paragraph. It is suggested to revise all through the manuscript.

6-     The introduction section is too long. Kindly shorten it.

7-     Authors tried to present the structure of the article at the end of the introduction. However, it is not well structured. Kindly improve on that.

8-      The novelty and practical applicability of this study should be highlighted more in the introduction section. The introduction section could be improved. Author should refer to these articles and several others relating to the subject matter from reputable journals to enrich and enhance the introduction and other sections of the manuscript. These are:

·       https://doi.org/10.1016/j.cscm.2021.e00783

·       https://doi.org/10.3390/polym13162610

·       https://doi.org/10.1016/j.matpr.2021.02.250

·       https://doi.org/10.1007/s42947-022-00224-4

·       https://doi.org/10.1007/978-3-031-26580-8_27

·       https://doi.org/10.3390/geotechnics3020009

9-      It is suggested to compare your findings with existing literature on different pre-treatment carried out on municipal sewage sludge. A comparison Table is expected between this work and other earlier related published works tabulating the latest works done in this domain to more effectively highlight the novelty of the present work. More explanations and interpretations must be added for the results.

10-   The study is novel and rich. However, it is surprising that no optimization was adopted using any of RSM, Uniform Design, Taguchi, or ANN?.

11-   There are several English language and coherence issues in the research. It requires proof reading and improvement. Text formatting and style needs to be consistent throughout the text.

12-   The conclusion is too long. It is suggested to re-organize the conclusion section much better. The conclusion is too much and lacks some basic components. Kindly summarize. It should be re-written in a well-structured manner. It should cover a summary of the problem(s), objectives methodology, findings, and recommendation(s).

13-   The reference style is not in line with the guideline provided in the journal template.

14-   The references cited are not enough to justify the work done in the manuscript. Kindly refer to the earlier suggested articles and several others to enrich the manuscript.

Finally, the article should be modified according to above-said comments and be thoroughly reviewed again before accepting it for publication.

There are several English language and coherence issues in the research. It requires proof reading and improvement. Text formatting and style needs to be consistent throughout the text.

Reviewer 4 Report

This manuscript tried to investigate the effect of utilizing metal waste strips on mechanical properties of road base materials. The issue is interesting and the manuscript is well written. There are some suggestions below to be applied before acceptance:

·         In the title, some of the words were written by capital letters and some of them with small ones. It is suggested to edit it based on the journal’s format.

·         Line 13 to 19 of the abstract can be summarized.

·         There are many updated research works which investigated the utilizing waste materials and fibers in constructional materials. It is suggested to refer them in the introduction. Some of them are suggested below:

o   Sukmak, P., Sukmak, G., De Silva, P., Horpibulsuk, S., Kassawat, S., & Suddeepong, A. (2023). The potential of industrial waste: Electric arc furnace slag (EAF) as recycled road construction materials. Construction and Building Materials368, 130393.

o   Kontoni, D. P. N., Onyelowe, K. C., Ebid, A. M., Jahangir, H., Rezazadeh Eidgahee, D., Soleymani, A., & Ikpa, C. (2022). Gene Expression Programming (GEP) Modelling of Sustainable Building Materials including Mineral Admixtures for Novel Solutions. Mining2(4), 629-653.

o   Segui, P., Safhi, A. E. M., Amrani, M., & Benzaazoua, M. (2023). Mining Wastes as Road Construction Material: A Review. Minerals13(1), 90.

·         It is suggested to add some figures of the road base materials too.

·         The test set-up should be clarified with adding a related figure or sketch.

·         The conclusion section can be summarized.

Round 2

Reviewer 1 Report

Dear authors,

I thank you for your efforts and the answers and corrections made.

With respect.

Reviewer 3 Report

Authors have significantly improved the manuscript.